# Recurrent horizontal transfer identifies mitochondrial positive selection in a transmissible cancer

Andrea Strakova ⓘ et al.#

Autonomous replication and segregation of mitochondrial DNA (mtDNA) creates the potential for evolutionary conflict driven by emergence of haplotypes under positive selection for 'selfish' traits, such as replicative advantage. However, few cases of this phenomenon arising within natural populations have been described. Here, we survey the frequency of mtDNA horizontal transfer within the canine transmissible venereal tumour (CTVT), a contagious cancer clone that occasionally acquires mtDNA from its hosts. Remarkably, one canine mtDNA haplotype, A1d1a, has repeatedly and recently colonised CTVT cells, recurrently replacing incumbent CTVT haplotypes. An A1d1a control region polymorphism predicted to influence transcription is fixed in the products of an A1d1a recombination event and occurs somatically on other CTVT mtDNA backgrounds. We present a model whereby 'selfish' positive selection acting on a regulatory variant drives repeated fixation of A1d1a within CTVT cells.

#A list of authors and their affiliations appears at the end of the paper.

Mammalian cells carry tens to thousands of copies of the mitochondrial genome (mtDNA), an autonomously replicating ~17,000 base-pair (bp) circular chromosome encoding genes essential for oxidative energy metabolism. Selection acts on genetic variation arising in mtDNA at two levels. First, it can act on cellular or organismal phenotypes produced by altered mitochondrial function (adaptive selection), and, second, it can drive changes in the frequency of haplotypes with altered replicative or segregation potential (selfish selection)[1]. The latter has been observed in several experimental and natural settings, and, if stronger than adaptive selection, has the potential to maintain or fix deleterious mitochondrial alleles[1–5]. Selfish selection may be particularly relevant in the context of therapeutic mitochondrial transplantation, and little is understood of the extent to which naturally occurring mtDNAs in human or animal populations vary in 'selfish' fitness traits[1,2,6,7].

One unusual but particularly valuable natural model for studying mtDNA dynamics is the canine transmissible venereal tumour (CTVT). CTVT is a contagious cancer in dogs, which spreads by the transfer of living cancer cells during mating, causing genital tumours[8]. The disease is found worldwide, and first arose from the somatic cells of an individual 'founder dog' that lived several thousand years ago[9,10]. Although the CTVT nuclear genome is clonal and represents the DNA of the founder dog, CTVT mtDNAs are polyclonal and were acquired periodically by horizontal transfer from transient hosts[11,12]. Capture of mtDNAs by CTVT cells results in a natural competition assay, whereby the relative fitness of diverse pairs of canine mtDNA haplotypes is assessed in vivo. Here, we identify repeated recent capture of a single canine mtDNA haplotype, A1d1a, by CTVT. This haplotype likely confers a selfish selective advantage via a regulatory polymorphism that may influence mtDNA transcription and replication.

## Results

**Recurrent horizontal transfer of the A1d1a haplotype**. We assessed the frequency of mtDNA capture in a cohort of 539 CTVT tumours collected from 43 countries across all inhabited continents (Supplementary Data 1). By comparing phylogenetic trees constructed using nuclear and mtDNA sequences (whole exomes[13] and full-length mtDNA genomes, respectively), we identified nineteen horizontal transfer events (Fig. 1a, Table 1, Supplementary Fig. 1, Supplementary Fig. 2, Supplementary Data 2). Remarkably, eleven of these mtDNA captures involved a single canine haplotype, known as A1d1a (Fig. 1a, Table 1, Supplementary Fig. 1, Supplementary Fig. 2, Supplementary Fig. 3, Supplementary Data 2)[14]. Despite the relatively high prevalence of A1d1a in the CTVT host dog germline population, it is evident that this haplotype is enriched for horizontal transfers to CTVT relative to other canine haplotypes ($p < 0.001$, empirical $p$ value derived from simulations, statistical significance was unchanged when restricting analysis to tumour–host pairs, Methods, Fig. 1b, Supplementary Fig. 4, Supplementary Data 1, Supplementary Data 3).

CTVT clusters derived from A1d1a horizontal transfers were observed in eight locations around the world, including Belize, Chile, Colombia, Grenada, India, Nicaragua, Paraguay and The Gambia (Fig. 1c, Table 1, Supplementary Fig. 4). The structure of the nuclear tree indicated that A1d1a replaced the two most common CTVT haplotypes, CTVT_HT1 and CTVT_HT2, on seven and four occasions, respectively (Fig. 1, Table 1, Supplementary Fig. 1). Ten of the eleven A1d1a horizontal transfers were fixed (homoplasmic) within CTVT cells; in the remaining A1d1a horizontal transfer (HT3, Table 1, Supplementary Data 1), A1d1a recombined with CTVT_HT1, possibly by copy choice

recombination[15], and the two tumours in the cohort derived from this horizontal transfer each carry a heteroplasmic mixture of mtDNA recombination products[12].

The accumulation of mtDNA somatic mutations (mutations that have arisen after each horizontal transfer event) provides an estimate of the time since mtDNA horizontal transfer[12]. Fourteen of nineteen horizontal transfer events in CTVT, including all eleven A1d1a captures, carry very few somatic mutations (mean 0.59 somatic mutations per mtDNA genome in A1d1a horizontal transfers, excluding variants whose germline or somatic status is unknown, Fig. 1d, Supplementary Data 4, Methods). Assuming a CTVT mtDNA somatic mutation rate of 0.0201 mutations per year (0.0127–0.0393, 95% highest posterior density interval[12,13], Methods), this implies that all A1d1a horizontal transfers occurred recently, probably within the last few decades (Fig. 1d, Supplementary Data 4).

It is possible that A1d1a may transfer between cells at a higher frequency than other haplotypes. To examine this possibility, we measured A1d1a copy number, a trait that may influence likelihood of mtDNA donation. MtDNA copy number did not differ between dog or tumour tissues with A1d1a and those with other haplotypes (Supplementary Fig. 5). Furthermore, dogs with germline A1d1a did not cluster on a nuclear phylogenetic tree (Supplementary Fig. 6), suggesting that it is unlikely that nuclear genetic features of A1d1a host dogs enable these animals to donate mtDNA to CTVT more efficiently. Although we cannot exclude the possibility that A1d1a does indeed transfer between cells at a higher frequency than other haplotypes, it seems more plausible that the underlying opportunity for horizontal transfer is constant between haplotypes, relative to population frequency. Once within the CTVT cell, traits encoded by A1d1a-specific variants may bestow a selective advantage relative to other haplotypes.

**Genetic features of the A1d1a haplotype**. We annotated the genetic features of A1d1a, relative to other canine haplotypes, in order to search for variation that may underlie this haplotype's selective advantage. A1d1a carries eight single-nucleotide polymorphisms (SNPs) and one short insertion that are unique to this haplotype (or, in some cases, shared with the related A1d1 haplotype) compared with sixteen other major dog haplotype groups (Fig. 2a, Supplementary Data 5). Six of these variants occur within protein-coding genes, one of which is predicted to cause a non-synonymous change (7593T>C in *MT-CO2*) (Fig. 2a, Supplementary Data 5). The remaining variants fall within ribosomal DNA (one variant) and within the mitochondrial control region (two variants) (Fig. 2a, Supplementary Data 5). The control region is a non-coding regulatory zone that contains elements required for the initiation of mtDNA replication and transcription[16,17]. Transfer event HT3, in which A1d1a and CTVT_HT1 have undergone recombination, provides an opportunity to further refine the candidate region for positive selection within A1d1a. Using long sequence reads, we phased recombinant mtDNA haplotypes within the two tumours derived from HT3 (Methods[12]). Both HT3 tumours carry a heteroplasmic pool of recombinant haplotypes; however, an insertion of two cytosines in the control region, 16660insCC, was the only A1d1a-specific variant carried by all haplotypes within these tumours (Fig. 2b). In addition, 16660insCC was twice observed arising as a somatic mutation, on CTVT_HT1 and CTVT_HT2 mtDNA backgrounds, respectively (Fig. 2a, Supplementary Data 5). Together, these findings present 16660insCC as a candidate for driving A1d1a-positive selection.

16660insCC occurs between conserved sequence block 3 (CSB3) and *tRNA-Phe* within the canine control region (Fig. 2c).

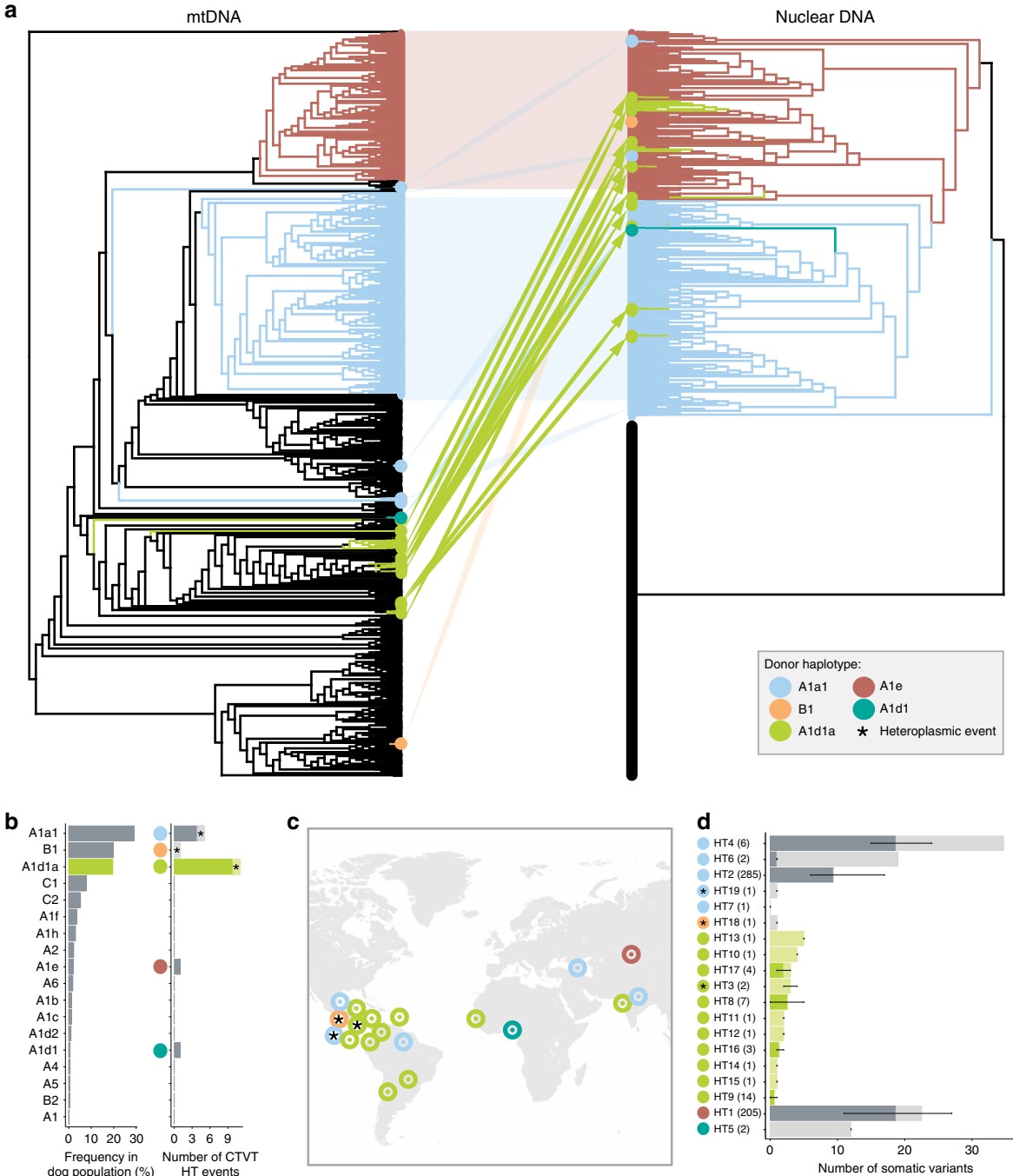

**Fig. 1 Recurrent and recent horizontal transfer of canine mitochondrial haplotype A1d1a. a** MtDNA (left) and nuclear DNA (right) maximum likelihood phylogenetic trees for 539 CTVT tumours (coloured) and 494 dogs (black). Correspondence between equivalent CTVTs on mtDNA and nuclear trees is indicated and coloured by mtDNA donor haplotype. Trees are presented as cladograms without informative branch lengths. High resolution trees are presented in Supplementary Figs. 1 and 2. **b** Frequencies of 18 canine mtDNA haplotypes within a representative global CTVT host dog population ($n =$ 495) (left), and the number of CTVT horizontal transfer (HT) events involving each haplotype in a population of 539 CTVTs (right). Donor haplotype (coloured dots) and heteroplasmic horizontal transfer events (asterisks (*) and lighter shading) are shown. Heteroplasmic tumours carry both the parental and introduced haplotype. The A1d1a heteroplasmic horizontal transfer event involves mtDNA recombination. Bars representing A1d1a are highlighted in green. **c** Inferred geographical locations of 19 CTVT mtDNA horizontal transfer events. Each horizontal transfer is represented by a dot coloured by donor haplotype. If all CTVTs arising from a horizontal transfer were sampled at the same location, then this was inferred as the location of the horizontal transfer. If CTVTs derived from the horizontal transfer were found in several locations, then the likely site of the horizontal transfer was inferred based on phylogenetic information[12,13]. Heteroplasmic horizontal transfer events are indicated with an asterisk (*). **d** Number of somatic mtDNA mutations acquired since each horizontal transfer (HT) event. Number of CTVTs ($n = 539$) belonging to each HT event is indicated. Bars are split into two categories: darker colour shades represent confident somatic mutations that are polymorphic within each HT group, with error bars representing the mutation range; lighter colour shades represent variants that are fixed within each HT group, whose somatic or germline status cannot be determined (see Methods). Bars representing A1d1a HTs are highlighted in green. Donor haplotype (coloured dots) and heteroplasmy (asterisk, *) are shown.

**Table 1 Summary of 19 horizontal transfer (HT) events detected in a population of 539 CTVT tumours.**

| Horizontal transfer | Donor haplotype group | CTVT haplotype group replaced | Number of tumours | Percentage heteroplasmy (%) | Locations observed |
|---|---|---|---|---|---|
| HT1 | A1e | Unknown | 205 | 100 | Widespread |
| HT2 | A1a1 | A1e (HT1) | 285 | 100 | Widespread |
| HT3 | A1d1a | A1e (HT1) | 2 | Recombinant | Nicaragua |
| HT4 | A1a1 | Unknown | 6 | 100 | India |
| HT5 | A1d1 | A1a1 (HT2) | 2 | 100 | Nigeria |
| HT6 | A1a1 | A1e (HT1) | 2 | 100 | Armenia |
| HT7 | A1a1 | A1e (HT1) | 1 | 100 | Mexico |
| HT8 | A1d1a | A1e (HT1) | 7 | 100 | Belize |
| HT9 | A1d1a | A1e (HT1) | 14 | 100 | Nicaragua |
| HT10 | A1d1a | A1e (HT1) | 1 | 100 | Nicaragua |
| HT11 | A1d1a | A1e (HT1) | 1 | 100 | Colombia |
| HT12 | A1d1a | A1e (HT1) | 1 | 100 | Colombia |
| HT13 | A1d1a | A1e (HT1) | 1 | 100 | Chile |
| HT14 | A1d1a | A1a1 (HT2) | 1 | 100 | The Gambia |
| HT15 | A1d1a | A1a1 (HT2) | 1 | 100 | Grenada |
| HT16 | A1d1a | A1a1 (HT2) | 3 | 100 | India |
| HT17 | A1d1a | A1a1 (HT2) | 4 | 100 | Paraguay |
| HT18 | B1 | A1e (HT1) | 1 | ~55 | Nicaragua |
| HT19 | A1a1 | A1e (HT1) | 1 | ~15 | Nicaragua |

Donor haplotype group, incumbent CTVT haplotype that was replaced, number of tumours observed within each horizontal transfer group, percentage heteroplasmy (the fraction of mtDNAs in the tumour derived from the incoming haplotype) and geographical locations in which tumours were observed are listed for each HT event. The two tumours derived from HT3 each carried a heteroplasmic mixture of recombinant haplotypes from the incumbent and incoming mtDNAs.

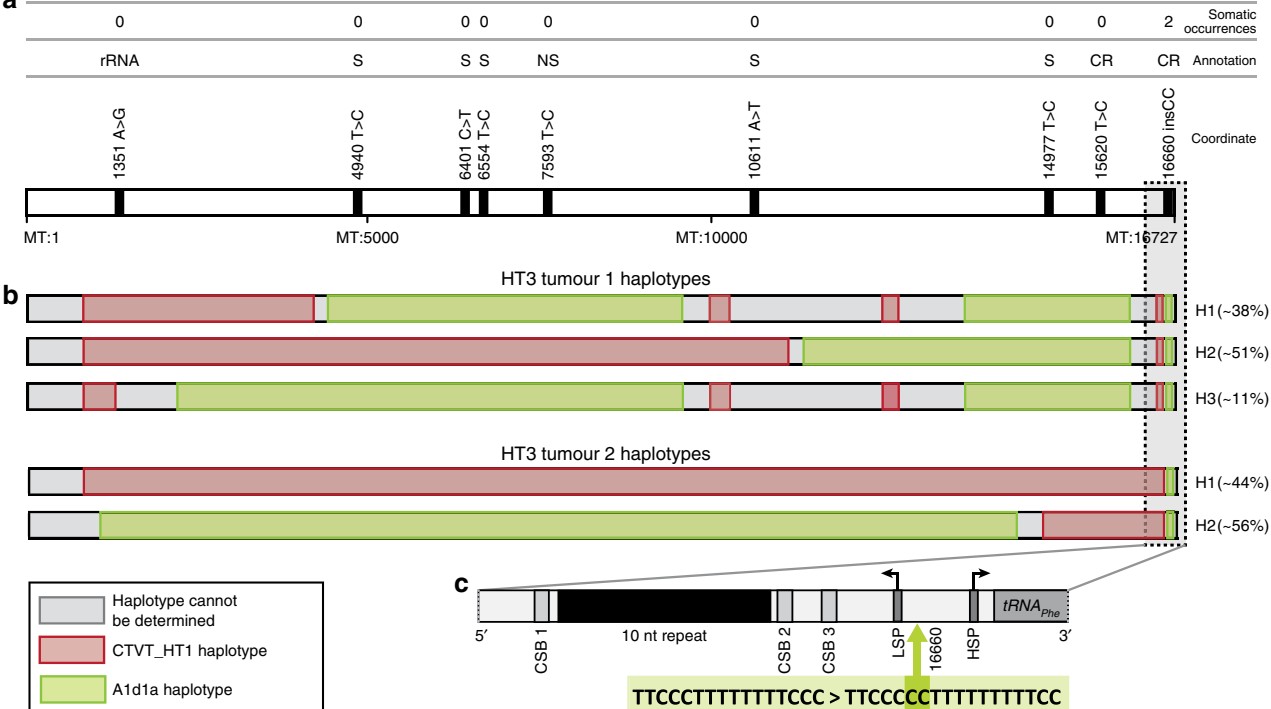

**Fig. 2 Genetic features of the A1d1a haplotype. a** Genetic variants unique to the A1d1a haplotype (and in some cases shared with the related A1d1 haplotype, Supplementary Data 5) relative to 16 other dog haplotype groups. Number of somatic occurrences on other non-A1d1a CTVT mtDNA backgrounds is indicated above each variant. Variants within protein-coding genes are annotated as synonymous (S) or non-synonymous (NS), and other variants are annotated as occurring within ribosomal RNA (rRNA) or the control region (CR). MtDNA (MT) genome coordinates are indicated. **b** MtDNA haplotypes detected using long-read sequencing in the two tumours (labelled tumour 1 and tumour 2) belonging to the HT3 group, in which A1d1a and CTVT_HT1 have undergone recombination, with a heteroplasmic mixture of recombination products present in each tumour. The region fixed in all haplotypes in both tumours is indicated with a dotted box. The estimated frequency of each recombinant haplotype (H) within the two tumours' CTVT cell mtDNA population is shown. **c** 16660insCC sequence context and position relative to control region features. Conserved sequence blocks (CSB) 1–3 are marked, together with the light strand promoter (LSP), heavy strand promoter (HSP) and a ten-nucleotide (nt) repeat block. 16660insCC co-occurs in A1d1a with 16672C>T, a polymorphism present on several canine haplotypes.

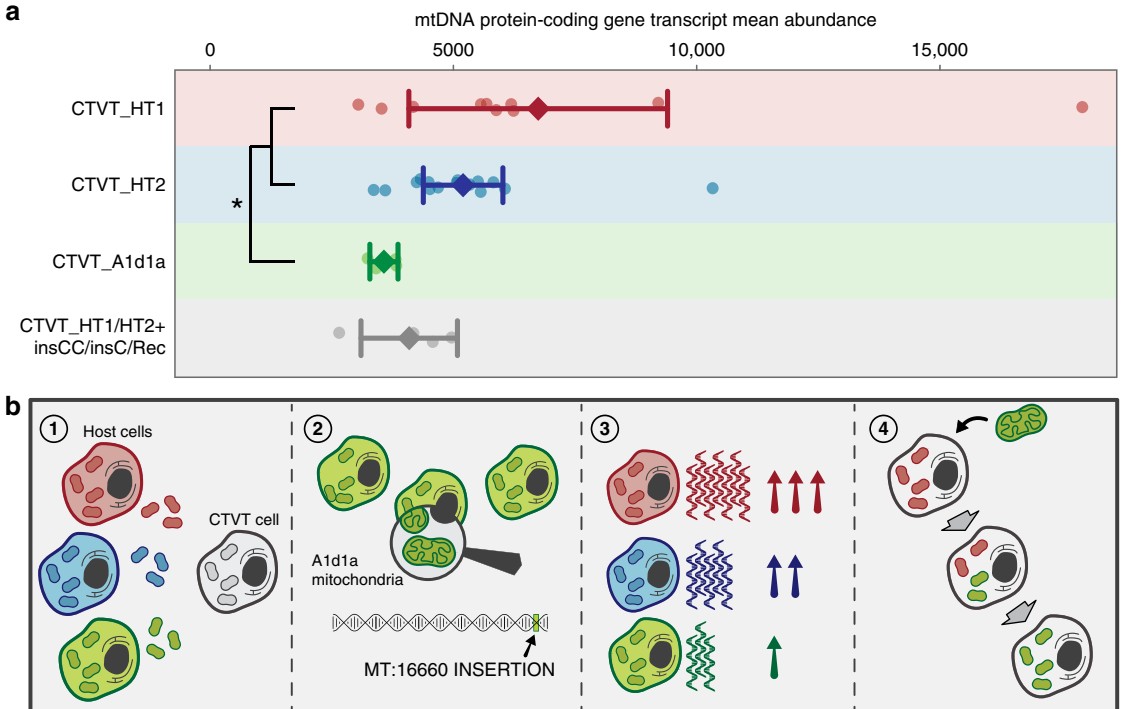

**Fig. 3 MtDNA gene transcript abundance and model. a** Mean abundance of mtDNA protein-coding gene RNA in CTVT_HT1, CTVT_HT2, CTVT_A1d1a and CTVT_HT1/HT2+insCC/insC/Rec CTVTs (n = 33). In the group of CTVT_HT1/HT2+insCC/insC/Rec CTVTs (represented in grey), two carry 16660insCC and one carries 16660insC as a somatic mutation on CTVT_HT1 or CTVT_HT2 haplotype backgrounds, and one carries 16660insCC on a CTVT_HT1/A1d1a recombinant background. Dots represent mean abundances of mtDNA protein-coding gene transcripts in each of the CTVTs from each group, diamonds indicate mean mtDNA protein-coding gene abundances for each group and bars indicate 95% confidence intervals of the mean. Asterisk (*), p = 0.015 (two-sided Mann–Whitney test; Methods). Supplementary Data 6B provides batch-corrected transcript abundance data. **b** A model explaining repeated CTVT capture of the A1d1a haplotype. (1) All canine mtDNA haplotypes have equal opportunity for CTVT horizontal transfer, relative to population haplotype frequency. (2) A1d1a mtDNA haplotype carries an insertion in the control region, 16660insCC, which is not present in other haplotypes, and which may have regulatory functions. (3) MtDNA protein-coding transcript abundance is decreased in tumours with the A1d1a mtDNA haplotype relative to the two most common CTVT mtDNA haplotypes. (4) After A1d1a mtDNA horizontal transfer, A1d1a becomes homoplasmic through a selfish replicative advantage.

Although this locus is poorly conserved across species, the equivalent sequence region within human mtDNA contains one of the two mitochondrial transcription promoters, the light strand promoter (LSP). Transcription initiation from LSP produces both the light strand polycistron and the primer for leading strand mtDNA replication initiation[18,19]. Using 5′ rapid amplification of cDNA ends, we mapped the canine LSP transcriptional start site (LSP-TSS) to position 16648, 12–15 bp downstream from 16660insCC (Methods). In humans, the transcription initiation complex recognises a region at an equivalent distance upstream of the LSP-TSS, suggesting that 16660insCC may lie within the transcription initiation complex binding module of canine LSP[20]. This finding implies that 16660insCC may influence transcriptional activity.

**Mitochondrial gene transcript abundance in A1d1a CTVT.** In order to further characterise the effects of 16660insCC on mtDNA transcription, we performed RNA sequencing on CTVT tumours carrying the two most abundant CTVT haplotypes CTVT_HT1 and CTVT_HT2, as well as those with A1d1a. The A1d1a CTVTs were phylogenetically unrelated on the nuclear tree, and were derived from three different horizontal transfer events (Supplementary Fig. 1, Supplementary Data 1, Supplementary Data 6A). Abundance of mtDNA protein-coding gene RNA, including heavy and light strand transcripts, was reduced in A1d1a CTVTs compared with CTVT_HT1 and CTVT_HT2

CTVTs (39% decrease in mean transcript abundance; p = 0.015, two-sided Mann–Whitney test), although there was large variation in transcript abundance within groups (Fig. 3a, Supplementary Data 6B). Mean mtDNA transcript abundance was also decreased in a group of tumours carrying 16660insCC (two tumours) and 16660insC (one tumour) somatic mutations on CTVT_HT1 and CTVT_HT2 haplotype backgrounds, and carrying 16660insCC on a recombinant CTVT_HT1/A1d1a background (one tumour), although this difference was not statistically significant (30% decrease in mean transcript abundance; p = 0.093, two-sided Mann–Whitney test) (Fig. 3a, Supplementary Data 6B). Furthermore, 7 of 13 mtDNA genes showed significantly lower abundance in A1d1a CTVTs compared with CTVT_HT1 and CTVT_HT2 CTVTs (q < 0.05, two-sided Wald test with multiple-testing correction; Methods) (Supplementary Data 6B). Abundance of a set of nuclear-encoded mitochondrially relevant gene transcripts was unchanged between CTVTs with A1d1a and those with CTVT_HT1 and HT2 haplotypes (Supplementary Data 6C). Introducing a CC insertion into an equivalent position to canine 16660 within the human LSP abolished light strand transcription in an in vitro assay (Supplementary Fig. 7), indicating that the locus is highly relevant for light strand transcriptional initiation. These analyses implicate 16660insCC as a regulatory variant acting to reduce mtDNA gene transcription.

Negative selection operating on mtDNA-encoded protein-coding genes indicates that mitochondrial function is important

for CTVT biology[12]. Thus, a decline in mtDNA gene transcript abundance, as observed in A1d1a CTVT, is unlikely to be adaptive for CTVT cells, although it may be neutral[21]. Nevertheless, we quantified two possible adaptive phenotypes, mitotic rate and timing of tumour response to chemotherapy. These traits did not differ between A1d1a CTVTs and CTVT_HT1 and CTVT_HT2 CTVTs (Supplementary Fig. 8). Furthermore, the set of A1d1a-specific polymorphisms does not immediately suggest a mechanism for adaptive functional relevance (Supplementary Data 5). Overall, we cannot exclude the possibility that A1d1a is adaptive for CTVT cells; however, there is no evidence supporting this hypothesis. Rather, our work supports a model whereby a regulatory variant in the A1d1a control region reduces transcription from this haplotype. Given that transcription and replication are coupled in mtDNA[22,23], we suggest that A1d1a favours replication over transcription, conferring A1d1a with a replicative advantage that drives this haplotype to fixation via selfish selection (Fig. 3b).

## Discussion

A survey of CTVT mtDNA diversity revealed that a single canine mtDNA haplotype, A1d1a, has repeatedly and recently colonised CTVT cells via horizontal transfer. Although we cannot rule out the possibilities that A1d1a has an increased propensity for horizontal transfer, or that it offers an adaptive advantage to CTVT cells, our data most strongly support a model whereby A1d1a replaces other mtDNA haplotypes via selfish selection.

MtDNA integrity is important for CTVT biology[12], and it has been proposed that mtDNA exchange per se is under positive selection via replacement of haplotypes degraded by somatic mutation[11,12]. However, assuming that A1d1a is horizontally transferred at a similar rate to other haplotypes, relative to population frequency, our observations suggest that most horizontal transfers result in maintenance of incumbent CTVT mtDNAs, despite their high somatic mutation burden. Perhaps centuries of haplotype replacement and intracellular haplotype competition have optimised 'selfish' traits within CTVT mtDNAs, such that few dog haplotypes may now compete with incumbent CTVT mtDNAs, despite a potential adaptive selective advantage. On the other hand, given its reduced mtDNA transcript abundance, it is plausible that A1d1a is in fact maladaptive for CTVT cells, highlighting a possible unexpected deleterious consequence of long-term clonal evolution.

Without knowledge of the underlying frequency of mtDNA horizontal transfer into CTVT, or of the number of mtDNA copies transferred per event, the strength of selection operating on A1d1a cannot be quantified. Furthermore, the lack of laboratory tools for studying canine mitochondrial replacement, in particular the unavailability of CTVT cell lines, precludes direct experimental investigation of this process. However, the observation of homoplasmic somatic 16660insCC mutations occurring on non-A1d1a CTVT mtDNAs raises the possibility that selection operating on this variant may, in some cases, be sufficient to drive fixation from a single molecule (although we cannot exclude the possibility that the observed somatic 16660insCC mutations achieved homoplasmy via neutral genetic drift).

Interestingly, most (14 of 19) horizontal transfer events observed in this CTVT cohort, including all eleven A1d1a horizontal transfers, occurred recently. This may reflect an exponentially expanding CTVT population[13], or, perhaps, a failure of ancient A1d1a CTVT lineages to persist. However, the frequency of mtDNA horizontal transfer has been linked to cell stress in a variety of model systems[24,25], and chemotherapy exposure has been proposed to increase the rate of mtDNA capture[26]. Thus, it

is possible that the use of chemotherapy, a widespread treatment for CTVT since the 1980s[27,28], has elevated the frequency of mtDNA capture in CTVT. Veterinarians in all locations where we observed recent mtDNA horizontal transfers reported chemotherapy use for CTVT. Furthermore, in some cases, veterinarians indicated that animals undergoing chemotherapy were not routinely isolated from other dogs, creating conditions that would permit transmission of chemotherapy-exposed CTVT cells. Varying access to and attitudes towards chemotherapy treatment in different cultural settings may also explain the apparent over-representation of horizontal transfer events in Central America (Fig. 1c), although we cannot exclude the possibility that this arose by chance. Thus, it is plausible that human intervention may have influenced the recent emergence of A1d1a CTVT clusters.

The replicative autonomy of mtDNA provides scope for selfish haplotype evolution, despite potential deleterious consequences for cellular or organismal fitness[3,4,29]. Such evolutionary conflicts are likely controlled in the germline via safeguarding strategies imposed during mtDNA bottlenecks in the egg[1,30,31]. Nevertheless, it has been suggested that these processes may drive somatic and germline persistence of 'selfish' mitochondrial alleles, some of which may be maladaptive or even pathogenic[2,30]. Importantly, the replicative drive of donor and recipient mtDNA haplotypes should be considered in mitochondrial transplantation therapy, and may be crucial for success of this procedure[1,6,7,32]. Although CTVT is considered a biological oddity, its periodic uptake and juxtaposition of mitochondrial haplotypes provides broad and unexpected insights into the evolutionary consequences of natural polymorphism in mammalian mitochondrial genomes.

## Methods

**Sample collection and nucleic acid extraction.** This study was approved by the Department of Veterinary Medicine, University of Cambridge, Ethics and Welfare Committee (reference number CR174). Informed consent for sample collection was received from the owner, or from another responsible person in the case of unowned dogs. Tumour and host (gonad, skin, blood or liver) tissue samples were collected into RNAlater (Invitrogen, Carlsbad, CA, USA) and stored at −20 °C until processing. Genomic DNA was extracted using the Qiagen DNeasy Blood and Tissue extraction kit (Qiagen, Hilden, Germany). Total RNA was extracted using the Qiagen AllPrep extraction kit (Qiagen, Hilden, Germany). Sample information is presented in Supplementary Data 1.

**Confirmation of CTVT diagnosis.** Quantitative PCR assays were performed to confirm CTVT diagnosis by detection of the CTVT-specific LINE-MYC genomic rearrangement[12], or by histopathology analysis.

**Exome library preparation and sequencing.** Illumina whole genome DNA sequencing libraries with insert size 100–400 base pairs (bp) were constructed using standard methods according to the manufacturer's instructions. Probes targeting the canine protein-coding genome and microRNAs were designed (43 Mb, Agilent Technologies, Santa Clara, CA, USA; ELID IDs 0679931, 0679881, 0679921, 0679911, 0679901 and 0679891), and used to capture the canine exome using standard methods according to the manufacturer's instructions. Sequencing was performed with 75-bp paired-end reads on an Illumina HiSeq2000 instrument, using V3 and V4 sequencing chemistry (Illumina, San Diego, CA, USA). Tumours and hosts were sequenced to an average exome depth of ~132× and ~104×, respectively. Sequence reads were aligned to the CanFam3.1 canine reference genome[33,34] using the Burrows–Wheeler Aligner's Backtrack Alignment[35] (BWA-backtrack) v0.5.9-r16+rugo with options '-l 32 -t 6'. Information on the genomic composition of the sequenced exome regions is detailed in Baez-Ortega et al.[13].

**RNA sequencing.** Stranded RNA sequencing libraries with insert size 100–300 bp were generated for 33 CTVTs (1208Ta-Dog, 1210Ta-Dog, 1247Ta-Dog, 126Ta-Dog, 131Ta-Dog, 1532Ta-Dog, 24Ta-Dog, 335Ta-Dog, 341Ta-Dog, 355Ta-Dog, 365Tb-Dog, 366Ta-Dog, 410Ta-Dog, 423Ta-Dog, 439Tb-Dog, 459Ta-Dog, 464Ta-Dog, 468Ta-Dog, 550Ta-Dog, 556Tb-Dog, 559Ta-Dog, 560Ta-Dog, 608Ta-Dog, 609Ta-Dog, 645Ta-Dog, 652Ta-Dog, 666Ta-Dog, 683Ta-Dog, 773T1a-Dog, 79Ta-Dog, 809Ta-Dog, 851Ta-Dog and 855Ta-Dog) with the Ribo-Zero ribosomal RNA removal kit (Illumina, San Diego, CA, USA) using standard methods according to the manufacturer's instructions. The libraries were sequenced with 75-bp paired-end reads on an Illumina HiSeq4000 instrument (Illumina, San Diego, CA, USA) to

an average depth of ~168×[13]. Samples were processed in three batches (Supplementary Data 6A).

**Substitution calling and post processing.** Substitutions in nuclear genomes were called using Platypus[36] v0.8.1 as part of a bespoke computational pipeline (Somatypus v1.3, https://github.com/baezortega/somatypus)[13]. Substitutions in mitochondrial genomes were called using the same bespoke computational pipeline as above, with the following modifications in the individual filtering and merging step, and in the final filtering step: substitution called if ≥3 supporting reads with mapping quality ≥ 20 and base quality ≥ 20 were detected, and base qualities within 10 bp of the ends of reads were set to 0.

**Indel calling.** Small insertions and deletions (indels) in mitochondrial genomes were extracted from whole exome sequencing data using samtools and bcftools[37,38]. Samtools v0.1.18 mpileup with options '-C50' and '-gf' was used to create raw bcf files from indexed bam files, and indels were called with bcftools v0.1.17 view, options '-bvcg'.

**Phylogenetic analyses.** For maximum likelihood topology inference, the mitochondrial DNA (mtDNA) phylogenetic tree represented in Fig. 1a and Supplementary Fig. 2 was constructed with 539 CTVTs, 494 dogs (Supplementary Data 1 and 2), the CanFam3.1 canine mitochondrial reference sequence, and rooted with two coyote genomes (GenBank accession numbers: DQ480510.1, DQ480509.1). We phased and extracted distinct haplotypes in heteroplasmic tumours using variant allele fractions or, in tumours belonging to HT3, using PacBio sequencing (see section 'MtDNA recombination analysis'), and each haplotype was included in the tree as a separate sample (Supplementary Data 2). One dog sample (15 Ha, Supplementary Data 1) was discarded due to CTVT contamination. The tree inference was performed with RAxML[39] v8.2.9 using a maximum likelihood method with a generalised time-reversible (GTR) substitution model. In total, 500 bootstrap replicates were produced using the rapid bootstrapping algorithm implemented in RAxML v8.2.9.

The nuclear DNA maximum likelihood phylogenetic tree represented in Fig. 1a and Supplementary Fig. 1 was constructed using 148,030 nuclear CTVT somatic mutations genotyped across the same set of 539 CTVT tumours and 494 normal dogs (Supplementary Data 1), using a GTR substitution model with a Gamma model of site heterogeneity in RAxML[13,39]. In total, 500 bootstrap replicates were produced using the rapid bootstrapping algorithm implemented in RAxML v8.2.9.

For tanglegram construction, correspondence between equivalent CTVT tumours on mtDNA and nuclear trees was assessed using a tanglegram constructed in Dendrocope[40] v3.5.10.

For horizontal transfer group inference, mtDNA horizontal transfer was inferred if discordance was observed between mtDNA and nuclear phylogenetic trees. Heteroplasmic horizontal transfer was inferred in tumours carrying two or more distinct mtDNA haplotypes, all present at more than 10% frequency, and not explained by the matched host haplotype (Table 1, Supplementary Data 2). Heteroplasmic tumours carried haplotypes from the parental CTVT and the newly transferred haplotype, except for the HT3 group, which carried a heteroplasmic mixture of recombinant haplotypes. Groups of CTVTs were inferred to belong to a single horizontal transfer event if they clustered together on both the mtDNA and nuclear phylogenetic trees (Fig. 1a, Supplementary Figs. 1 and 2).

**Dog haplotype assignment.** The haplotype naming system was adapted from the cladistic canine mtDNA phylogeny nomenclature proposed by Fregel et al.[14]. An mtDNA phylogenetic tree was constructed with 495 dogs (Supplementary Data 1), the CanFam3.1 canine mitochondrial reference sequence, and rooted with two coyotes (GenBank accession numbers: DQ480510.1, DQ480509.1) (Supplementary Fig. 3). The tree inference was performed with RAxML[39] v8.2.9 using a maximum likelihood method with a GTR substitution model. The tree topology was used to assign dogs into one of the following 18 haplotypes: A1, A1a1, A1b, A1c, A1d1, A1d1a, A1d2, A1e, A1f, A1h, A2, A4, A5, A6, B1, B2, C1 and C2 (Fig. 1b, Supplementary Fig. 3, Supplementary Data 3).

**Timing analysis of horizontal transfer events.** CTVT_HT1 and CTVT_HT2 diverged ~469.28 years ago (240.34–744.31, 95% highest posterior density interval)[13]. Given that the mean number of CTVT_HT2 somatic mtDNA mutations = 9.437, and assuming a constant mtDNA mutation rate, we estimate a CTVT mtDNA mutation rate of 0.0201 mutations per year (0.0127–0.0393, 95% highest posterior density interval) (Supplementary Data 4). The number of substitutions in horizontal transfer events HT1, HT2, HT4 and HT5 is presented as in Strakova et al.[12]. In HT3, somatic or germline status of substitutions could not be determined due to mtDNA recombination.

**A1d1a horizontal transfer frequency modelling.** Simulations were performed to calculate the probability of A1d1a horizontal transfer occurring at the frequency observed, given the observed frequency of haplotypes in the dog population (Supplementary Data 3), and the assumption that each haplotype has an equal horizontal transfer opportunity. The simulation was performed in R[41] over 10000

repetitions, with randomisation seed set to 76. The empirical *p* value was derived from the proportion of simulations with the number of A1d1a horizontal transfers equal to or exceeding the observed value. A more conservative analysis was performed that included only matched tumour–host pairs. The code used for this analysis is available at https://github.com/TransmissibleCancerGroup/Mixed.

**A1d1a mtDNA copy number.** MtDNA copy number was calculated for normal canine ovarian ($n = 42$) and testicular ($n = 41$) tissue, as well as for CTVT tumours ($n = 337$) (Supplementary Fig. 5). The following equation was used to calculate the mtDNA copy number: (mtCOV-T/nuclCOV-T) × P, where mtCOV-T = average coverage across the mitochondria × mtDNA tumour fraction, nuclCOV-T = average coverage across the nuclear genome × nuclear tumour fraction, and P = ploidy. Ploidy was estimated as two for both CTVT tumours and dogs[42]. MtDNA tumour fraction was estimated from variant allele fraction (VAF, i.e. number of reads supporting the substitution variant as a fraction of the total number of reads covering the substitution variant position). Nuclear tumour fraction was estimated from whole exome sequencing data using the following equation: (mode of T-VAF) × 2, where T-VAF = somatic tumour variant allele fraction[13]. Statistical significance was tested using an unpaired two-tailed Student's *t* test implemented in R[41].

**Nuclear relationships of A1d1a dogs.** A neighbour-joining cladogram was constructed using 136,880 canine germline variants genotyped across 495 CTVT host dogs (Supplementary Data 1) and 7 wolves obtained from the Dog Genome SNP Database (ftp://download.big.ac.cn/dogsd/bam/)[43] (Sample IDs: ISW, CRW, CHW, LUPWRUS00001, LUPWRUS00002, LUPWRUS00003 and LUPWCHN00001) (Supplementary Fig. 6)[13]. To account for the presence of heterozygous and homozygous variants, distances between the aligned sequences were calculated using the p-distance metric, as implemented in the 'dist.p' function of the phangorn[44] v2.3.1 package for R. The phylogenetic tree was constructed using the neighbour-joining algorithm, as implemented in the 'NJ' function of the phangorn package.

**A1d1a-specific substitutions and indels.** Substitutions and small insertions and deletions (indels), which were common and unique to A1d1a and, in some cases, the related A1d1 haplotype, were identified through filtering against other CTVT and dog samples (CTVTs belonging to HT3, which has undergone recombination between CTVT_HT1 and A1d1a, were not considered in defining A1d1a-specific variants) (Supplementary Data 1 and 5). Indels were validated visually using Integrative Genomics Viewer (IGV)[45,46]. Annotation was performed using Variant Effect Predictor[47] (Supplementary Data 5), and the functional relevance of each variant was assessed, where possible, by aligning the region to humans and assessing variants using Mitomap[48]. Somatic occurrence of each A1d1a-specific genetic variant was assessed in the set of 495 CTVT tumours (Supplementary Data 1 and 5).

**MtDNA recombination analysis.** Potential recombinant samples were identified by searching for outliers on the mtDNA phylogenetic tree, and by visualising mtDNA VAF plotted against mtDNA genome coordinate. Standard PacBio genomic libraries were created using 5 μg of genomic DNA from samples 559T and 1315T, belonging to HT3, not utilising shearing or amplification techniques[12]. PacBio (Pacific Biosciences, Menlo Park, CA, USA) data reads aligned to CanFam3.1 mtDNA genome were viewed in SMRT view v2.3.0 (Pacific Biosciences, Menlo Park, CA, USA) and in IGV[45,46]. The three or two most common haplotypes in 559T and 1315T, respectively, were phased by visual inspection, as shown in Fig. 2b. Additional haplotypes present at a low level (less than 5%) were also identified in both samples, and are not shown in Fig. 2b.

**Gene expression analysis.** Two FASTQ files containing raw forward and reverse reads, respectively, were generated from each tumour BAM file using the biobambam2[49] v2.0.79 software, and gene expression was estimated via transcript abundance quantification using the Salmon[50] v0.8.2 software[13]. Gene-specific differential expression analysis was performed using the DESeq2[51] v1.14.1 R package to compare CTVT_A1d1a expression against CTVT_HT1 and CTVT_HT2 expression (Supplementary Data 6). To account for batch effects, both the batch and HT group of each sample were included as variables in the design formula for the differential expression analysis. Because the analysis was directed to mtDNA-encoded genes ($n = 13$, Supplementary Data 6B) and nuclear-encoded mitochondrially relevant genes ($n = 7$, Supplementary Data 6C), restricted hypothesis testing was performed, such that only these genes were considered when adjusting *p* values via Benjamini–Hochberg multiple-testing correction.

For assessment of overall changes in transcript abundance between HT groups, raw read counts of mtDNA protein-coding genes were transformed using the rlog function in DESeq2 and subjected to batch effect correction using the limma[52] v3.38.3 R package. After batch effect correction, the log transformation was reversed, and the corrected estimates were scaled to render them comparable to the original transcript abundance estimates. Mean abundances of mtDNA protein-coding genes per sample and per HT group were calculated from the batch-corrected estimates (Fig. 3a, Supplementary Data 6B). Statistical significance of

overall transcript abundance differences between groups was tested using a two-sided Mann–Whitney test (as implemented in R[41]) on the batch-corrected mtDNA transcript abundances per gene in each group (not the mean abundances). To confirm the result from batch-corrected estimates, uncorrected mtDNA transcript abundance estimates for group CTVT_A1d1a and groups CTVT_HT1 and CTVT_HT2 were compared independently for batches 1 and 2, and transcript abundance was found to be comparably decreased in group CTVT_A1d1a for each batch (49% decrease and $p = 0.0002$ for batch 1; 46% decrease and $p = 0.088$ for batch 2; batch 2 had only one A1d1a CTVT).

**Canine LSP mapping.** The MDCK cell line (ECACC 84121903) was grown adherently in DMEM (31966047, Gibco) with penicillin–streptomycin (10,000 U/ml, Gibco) and 10% foetal bovine serum (10270106, Gibco). Mitoplast extract was isolated[53], and RNA was extracted using the miRNeasy Mini Kit (Qiagen, Hilden, Germany). The position of the LSP transcriptional start site (LSP–TSS) was mapped using 5′/3′ RACE kit, 2nd Generation (Roche, Basel, Switzerland), according to the manufacturer's instructions. Briefly, cDNA was synthesised using a pre-designed LSP-P1 primer (5′-GTA AAC TCA TGT CAT CTA TTA TAC-3′), purified and tailed using dATP and terminal transferase. A polymerase chain reaction (PCR) was performed using a specific LSP-P2 primer (5′-CTT ATT TAT GTC CCG CCA AAC C-3′) and an oligo dT-Anchor primer. The PCR product was gel-purified and sequenced to identify the LSP-TSS.

To identify the mtDNA haplotype of MDCK, DNA was extracted using the AllPrep extraction kit (Qiagen, Hilden, Germany), and the region between mitochondrial positions 16499-111 was amplified in a PCR reaction using the primers F-16499 (5′-CCC CGT AAA CTC ATG TCA TCT ATT-3′) and R-111 (5′-GTG GAG GCT TGC ATG TGT AA-3′). The PCR product was purified using the QIAquick kit (Qiagen, Hilden, Germany) and sequenced to confirm the presence of 16660insCC, indicating that MDCK has haplotype A1d1a. We confirmed that the LSP–TSS was unchanged in non-A1d1a canine cells by performing promoter mapping in a second canine cell line (canine fibroblast cell line grown using sample 135H) with haplotype B1.

**MtDNA transcription assays.** Recombinant human POLRMT, TFAM, TFB2M and TEFM were expressed and purified[54,55]. Templates for transcription reactions consisted of pUC18 containing human mtDNA sequence (positions 1–477 for LSP-containing templates and positions 1–742 for templates containing both LSP and HSP, heavy strand promoter), cloned between BamHI and HindIII restriction sites. Constructs were created to include a wild-type template, +CC template to model canine 16660insCC at the equivalent position in the human mitochondrial genome (i.e. ~15 bp upstream of LSP transcription start site) and +TT template to model 16660insCC, and to reflect the stretch of thymines in the human sequence. Templates were linearised with either BamHI or HindIII (as indicated, Supplementary Fig. 7) to generate the appropriate runoff transcription products. Transcription reactions (25 μl) contained 500 fmol POLRMT, 500 TFB2M, TFAM (as indicated, Supplementary Fig. 7), 90 fmol template DNA, 10 mM Tris-HCl (pH 8.0), 10 mM MgCl$_2$, 64 mM NaCl, 100 μg/ml BSA, 1 mM DTT, 100 μM ATP, 100 μM CTP, 100 μM GTP, 10 μM UTP, 0.02 μM [α-32P] UTP (3000 Ci/mmol) and 4 units of murine RNase inhibitor (New England Biolabs, Ipswich, MA, USA). Where indicated (Supplementary Fig. 7), reactions also contained 1 pmol TEFM. Reactions were incubated at 32 °C for 30 min, then stopped by the addition of 200 μl of stop buffer (10 mM Tris-HCl (pH 8.0), 0.2 M NaCl, 1 mM EDTA, 150 μg/ml proteinase K and 0.1 mg/ml glycogen) and incubated at 42 °C for 45 min. Reactions were then ethanol-precipitated and resuspended in 10 μl of loading buffer (98% formamide, 10 mM EDTA, 0.025% xylene cyanol FF and 0.025% bromophenol blue). Samples were separated by 4% denaturing PAGE (1× TBE, 7 M urea) and imaged using autoradiography.

Canine mtDNA transcription reactions were also attempted. MDCK mitoplast lysate was isolated[53]. Templates for transcription reactions consisted of pEX-A128 containing canine mtDNA sequence (positions 16430–16727 for templates assumed to contain both LSP and HSP). Constructs were created to include a non-A1d1a template and an A1d1a template containing 16660insCC. Templates were linearised with BamHI and NotI, or EcoRI and SspI to generate runoff transcription products of variable lengths. Transcription reactions (25 μl) were performed and analysed as described above, with varying amounts of freshly extracted MDCK mitoplast lysate (1.25, 2.5 μl) and varying amounts of template DNA (100 and 200 fmol). No transcription from canine LSP or HSP was detected using this method.

**Mitotic rate.** A blinded histopathology scoring study was performed by a single scorer, under the guidance of a veterinary pathologist. Twenty-eight A1d1a CTVT histology samples were scored, together with 31 CTVT_HT1 and CTVT_HT2 CTVT controls. CTVT_HT1 and CTVT_HT2 controls were phylogenetically matched with A1d1a tumours. The 28 A1d1a CTVTs belonged to 8 horizontal transfer groups (HT3, 2 tumours; HT8, 6 tumours; HT9, 13 tumours; HT10, 1 tumour; HT13, 1 tumour; HT14, 1 tumour; HT16, 1 tumour; HT17, 3 tumours). Histology slides were scanned using a Hamamatsu Nanozoomer 2.0-HT slide scanner (C9600) before scoring, and were viewed using the NDP.view2 software (Hamamatsu Photonics, Hamamatsu, Japan). Each field was defined to consist of at least half CTVT parenchyma, avoiding areas of artefactual distortion, ulceration or haemorrhage. The number of mitotic figures (including bizarre mitotic figures) per 10 high-power fields (40×, a mitotic box of area 0.0325 cm$^2$, perimeter 722 μm on NDP.view2) was assessed. Statistical significance was tested using an unpaired two-tailed Student's $t$ test implemented in R[41].

**CTVT response to vincristine chemotherapy treatment.** A blinded study following response to vincristine chemotherapy in 53 CTVT tumours (4 A1d1a CTVTs and 49 CTVT_HT1 and CTVT_HT2 CTVTs) was performed in Belize and Colombia (13 CTVTs in Belize, 40 CTVTs in Colombia). CTVT tumours were treated weekly with intravenous vincristine sulfate (dose 0.02–0.025 mg/kg or 0.5 mg/m$^2$), and the time in days taken for the tumour to reduce to 50% of its original volume was measured (Supplementary Fig. 8). A single tumour belonging to the CTVT_HT1 and CTVT_HT2 group that did not reach 50% of its original volume within the study period was excluded from this analysis. The mitochondrial haplotype of each tumour was determined retrospectively by genotyping. Statistical significance was tested using an unpaired two-tailed Student's $t$ test implemented in R[41].

**Reporting summary.** Further information on research design is available in the Nature Research Reporting Summary linked to this article.

## Data availability

Whole-exome sequence data have been deposited in the European Nucleotide Archive (ENA) under accession number ERP109580. PacBio long-read sequence data have been deposited in ENA under accession number ERP120021. Gene expression data have been deposited in ArrayExpress under accession number E-MTAB-9037. The remaining data supporting the findings of this study are available within the article and its supplementary information files.

## Code availability

The code used for the A1d1a horizontal transfer frequency modelling analysis is available at https://github.com/TransmissibleCancerGroup/Mixed.

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

## Acknowledgements

This work was supported by a grant from Wellcome (102942/Z/13/A) and by a Philip Leverhulme Prize from the Leverhulme Trust. A.St. was supported by a Postgraduate Student Award from the Kennel Club Charitable Trust, by an EMBO Short-Term Fellowship (7761) and by the Ruth Bowden Scholarship from the British Federation of Women Graduates. T.J.N. is the recipient of a Sir Henry Dale Fellowship jointly funded by the Wellcome Trust and the Royal Society (213464/Z/18/Z) and a Rosetrees and Stoneygate Trust Research Fellowship (M811). The work was also supported by grants from Swedish Research Council (M.Fal. and C.M.G.), Swedish Cancer Foundation (M.Fal. and C.M.G.), European Research Council (M.Fal.) and the Knut and Alice Wallenberg Foundation (M. Fal. and C.M.G.). P.F.C. is a Wellcome Trust Principal Research Fellow (212219/Z/18/Z), and a UK NIHR Senior Investigator, who receives support from the Medical Research Council Mitochondrial Biology Unit (MC_UP_1501/2), the Evelyn Trust and the National Institute for Health Research (NIHR) Biomedical Research Centre based at Cambridge University Hospitals, NHS Foundation Trust and the University of Cambridge. The views expressed are those of the authors and not necessarily those of the NHS, the NIHR or the Department of Health. We acknowledge the Core Sequencing Facility, IT groups and members of the Cancer Genome Project at the Wellcome Sanger Institute. We are grateful to Michael R. Stratton for institutional support. We thank Hansong Ma, Michal Minczuk, Robin Weiss, Maximilian R. Stammnitz, Young Mi Kwon and members of the Transmissible Cancer Group for helpful discussions. We appreciate the laboratory support from Bradley Peter, Hector Diaz and other members of the Gustafsson and Falkenberg laboratories. We thank the following individuals for useful information and for their help obtaining samples for this project: Juliana Alzate-Ocampo, Diana Argüello, Jose Ignacio Arias, Sue Barrass, Ekaterina Batrakova, Rafaela Bortolotti Viéra, Nikki Brown, Fernando Constantino Casas, John Cooper, Amici Cannis Cotacachi, Johan de Vos, Lytvynenko Dmytro, Phillip Farnham, Ariberto Fassati, Andres Fernandez-Riomalo, Ricardo Gaitan, David Hanzlíček, Rafael Ricardo Huppes, John M. Igundu, Matilde Jimenez-Coello, Debra Kamstock, Patrick Kelly, Anna Klucnika, Tatiana Korytina, Anna Kuznetsova, Gleidice Eunice Lavalle (Universidade Federal de Minas Gerais), Olakunle AbdulRasaq Lawal, Thabo Lerotholi, Marco Lima-Maigua, Jimmy Loayza-Feijoo, Mayra López-Bucheli, Mwangi Maina, Margarita Mancero-Albuja, Cynthia Marchiori Bueno, Luis Martínez-López, Alfredo Martínez-Meza, Talita Mariana Morata Raposo, Jude Mulholland, Claudio Murgia, Alvira Murison Swartz, Fran Nargi, Marsden M. Onsare, Edwin Ortiz-Rodríguez, Elisabeth Peach, Lisa Pellegrini, Gerry Polton, Freddy Proaño-Pérez, Cameron Raw, Ceseltina Semedo, Sanjay Singh, Ivan Stoikov, Mirela Tinucci Costa, Emily Turitto, M. Rifat Vural, David Walker, Kevin Xie, Maurice Zandvliet, staff at Animal Medical Centre Belize City (Belize), veterinary surgeons and staff at Help in Suffering (Jaipur, India), staff at Hopkins Belize Humane Society (Belize), veterinary workers at Pet Centre (UVAS, Lahore, Pakistan), students from St. George's University (True Blue, Grenada, West Indies) who assisted with sample collection, staff at Veterinary Clinic 'El Roble' (Chile), staff and volunteers at World Vets (Gig Harbor, USA) and staff at the WVS International Training Centre in Ooty (India). We are grateful to the following organisations for helpful information: American College of Veterinary Internal Medicine (ACVIM), Animal Balance, Animal Care Association (The Gambia), Animal Management in Rural and Remote Indigenous Communities (AMRRIC), Associacao Bons Amigos de Cabo Verde, Humane Society of Cozumel, Humane Society Veterinary Medical Association–Rural Area Veterinary Services (HSVMA–RAVS), Israel Veterinary Medical Association, Italian Veterinary Oncology Society, Rural Vets South Africa, Veterinary Cancer Society, Veterinary Society of Surgical Oncology (VSSO), VetPharma, Vets Beyond Borders, ViDAS and Coco's Animal Welfare, The Spanky Project, VWB/VSF Canada, West Arnhem Land Dog Health Programme (WALDHeP), World Small Animal Veterinary Association (WSAVA) and МИР ВЕТЕРИНАРИИ (World Veterinary Medicine). The map in Fig. 1c and Supplementary Fig. 4 was used under the Royalty Free Licence from Free Vector Maps.

## Author contributions

E.P.M. conceived and designed the study. A.St. contributed to the study design and performed the overall study. T.J.N., A.B.-O., M.N.L., A.T.S., I.A.G.B., K.G. and J.W. performed the laboratory work or computational analysis. K.H. oversaw the histopathology work. I.A.-O., J.L.A., K.M.A., C.L.A., L.B.-I., T.N.B., J.L.B., K.B., C.B., A.C.D., A.M.C., H.R.C., J.T.C., S.M.Cu., E.D., K.F.d.C., A.B.D.N., A.P.d.V., L.D.K., E.M.D., A.R.

E.H., I.A.F., M.Faz., E.F., S.N.F, F.G.-A., O.G., P.G.G., R.F.H.M, J.J.G.P.H., R.S.H., N.I., Y.K., C.K., D.K., A.K., S.J.K., M.L.-P., M.L., A.M.L.Q., T.L., G.M., S.M.Ca., M.F.M.-L., B.M.M, M.M., E.J.M., B.N., K.B.N., W.N., S.J.N., A.O.-P., F.P.-O., M.C.P., K.P., R.J.P., J.C.R.-A., J.F.R., J.R.G., H.S., S.K.S., O.S., A.G.S., A.E.S.-S., A.Sv., L.J.T.M., I.T.N., C.G.T., E.M.T., M.G.v.d.W., B.A.V., S.A.V., O.W., A.S.W.-M., S.A.E.W. and I.Z. provided the clinical samples. P.F.C., T.J.N., M.Fal. and C.M.G. provided conceptual advice. M.Fal. and C.M.G. oversaw the transcription assay analysis. A.St. and E.P.M. wrote the paper. All authors commented on and approved the paper.

## Competing interests

The authors declare no competing interests.

## Additional information

Andrea Strakova[1], Thomas J. Nicholls[2,3,68], Adrian Baez-Ortega[1,68], Máire Ní Leathlobhair[1], Alexander T. Sampson[1], Katherine Hughes[4], Isobelle A. G. Bolton[1], Kevin Gori[1], Jinhong Wang[1], Ilona Airikkala-Otter[5], Janice L. Allen[6], Karen M. Allum[7], Clara L. Arnold[8], Leontine Bansse-Issa[9], Thinlay N. Bhutia[10], Jocelyn L. Bisson[1], Kelli Blank[8], Cristóbal Briceño[11], Artemio Castillo Domracheva[12], Anne M. Corrigan[13], Hugh R. Cran[14], Jane T. Crawford[15], Stephen M. Cutter[6], Eric Davis[16], Karina F. de Castro[17], Andrigo B. De Nardi[18], Anna P. de Vos[19], Laura Delgadillo Keenan[20], Edward M. Donelan[6], Adela R. Espinoza Huerta[21], Ibikunle A. Faramade[22], Mohammed Fazil[23], Eleni Fotopoulou[24], Skye N. Fruean[25], Fanny Gallardo-Arrieta[26], Olga Glebova[27], Pagona G. Gouletsou[28], Rodrigo F. Häfelin Manrique[29], Joaquim J. G. P. Henriques[30], Rodrigo S. Horta[31], Natalia Ignatenko[32], Yaghouba Kane[33], Cathy King[7], Debbie Koenig[7], Ada Krupa[34], Steven J. Kruzeniski[21], Marta Lanza-Perea[13], Mihran Lazyan[35], Adriana M. Lopez Quintana[36], Thibault Losfelt[37], Gabriele Marino[38], Simón Martínez Castañeda[39], Mayra F. Martínez-López[40], Bedan M. Masuruli[41], Michael Meyer[42], Edward J. Migneco[43], Berna Nakanwagi[44], Karter B. Neal[45], Winifred Neunzig[7], Sally J. Nixon[46], Antonio Ortega-Pacheco[47], Francisco Pedraza-Ordoñez[48], Maria C. Peleteiro[49], Katherine Polak[50], Ruth J. Pye[51], Juan C. Ramirez-Ante[48], John F. Reece[52], Jose Rojas Gutierrez[53], Haleema Sadia[54], Sheila K. Schmeling[55], Olga Shamanova[56], Alan G. Sherlock[51], Audrey E. Steenland-Smit[9], Alla Svitich[57], Lester J. Tapia Martínez[21], Ismail Thoya Ngoka[58], Cristian G. Torres[59], Elizabeth M. Tudor[60], Mirjam G. van der Wel[61], Bogdan A. Viţălaru[62], Sevil A. Vural[63], Oliver Walkinton[51], Alvaro S. Wehrle-Martinez[64], Sophie A. E. Widdowson[65], Irina Zvarich[66], Patrick F. Chinnery[67], Maria Falkenberg[2], Claes M. Gustafsson[2] & Elizabeth P. Murchison[1✉]

[1]Transmissible Cancer Group, Department of Veterinary Medicine, University of Cambridge, Cambridge, UK. [2]Department of Medical Biochemistry and Cell Biology, University of Gothenburg, Gothenburg, Sweden. [3]Wellcome Centre for Mitochondrial Research, Newcastle University, Newcastle Upon Tyne, UK. [4]Department of Veterinary Medicine, University of Cambridge, Cambridge, UK. [5]Worldwide Veterinary Service, International Training Center, Tamil Nadu, India. [6]Animal Management in Rural and Remote Indigenous Communities (AMRRIC), Darwin, Australia. [7]World Vets, Gig Harbor, USA. [8]Hopkins Belize Humane Society, Hopkins, Belize. [9]Animal Shelter, Stichting Dierenbescherming Suriname, Paramaribo, Suriname. [10]Sikkim Anti-Rabies and Animal Health Programme, Department of Animal Husbandry, Livestock, Fisheries and Veterinary Services, Government of Sikkim, Sikkim, India. [11]ConserLab, Animal Preventive Medicine Department, Faculty of Animal and Veterinary Sciences, University of Chile, Santiago, Chile. [12]Corozal Veterinary Hospital, University of Panamá, Panama City, Republic of Panama. [13]St. George's University, True Blue, Grenada. [14]The Nakuru District Veterinary Scheme Ltd, Nakuru, Kenya. [15]Animal Medical Centre, Belize City, Belize.

[16]International Animal Welfare Training Institute, UC Davis School of Veterinary Medicine, Davis, USA. [17]Centro Universitário de Rio Preto (UNIRP), São José do Rio Preto, São Paulo, Brazil. [18]School of Agricultural and Veterinary Sciences, São Paulo State University (UNESP), Jaboticabal, Brazil. [19]Ladybrand Animal Clinic, Ladybrand, South Africa. [20]Veterinary Clinic Sr. Dog's, Guadalajara, Mexico. [21]World Vets Latin America Veterinary Training Center, Granada, Nicaragua. [22]National Veterinary Research Institute, Vom, Nigeria. [23]Animal Clinic, Mombasa, Kenya. [24]Intermunicipal Stray Animals Care Centre (DIKEPAZ), Perama, Greece. [25]Animal Protection Society of Samoa, Apia, Samoa. [26]Faculty of Veterinary Science, University of Zulia, Maracaibo, Venezuela. [27]Veterinary Clinic BIOCONTROL, Moscow, Russia. [28]Faculty of Veterinary Medicine, School of Health Sciences, University of Thessaly, Karditsa, Greece. [29]Veterinary Clinic El Roble, Animal Healthcare Network, Faculty of Animal and Veterinary Sciences, University of Chile, Santiago de Chile, Chile. [30]OnevetGroup, Hospital Veterinário Berna, Lisboa, Portugal. [31]Universidade Vila Velha, Vila Velha, Brazil. [32]Veterinary Clinic Zoovetservis, Kiev, Ukraine. [33]École Inter-états des Sciences et Médecine Vétérinaires de Dakar, Dakar, Senegal. [34]Department of Small Animal Medicine, Faculty of Veterinary Medicine, Utrecht University, Utrecht, The Netherlands. [35]Vetexpert Veterinary Group, Yerevan, Armenia. [36]Lopez Quintana Veterinary Clinic, Maldonado, Uruguay. [37]Clinique Veterinaire de Grand Fond, Saint Gilles les Bains, Reunion, France. [38]Department of Veterinary Sciences, University of Messina, Messina, Italy. [39]Facultad de Medicina Veterinaria y Zootecnia, Universidad Autónoma del Estado de México, Toluca, Mexico. [40]School of Veterinary Medicine, Universidad de las Américas, Quito, Ecuador. [41]Veterinary Council of Tanzania, Dar-es-Salaam, Tanzania. [42]Touray & Meyer Vet Clinic, Serrekunda, The Gambia. [43]Hillside Animal Hospital, St. Louis, USA. [44]The Kampala Veterinary Surgery, Kampala, Uganda. [45]Asavet Veterinary Charities, Tucson, USA. [46]Vets Beyond Borders, Bylakuppe, India. [47]Faculty of Veterinary Medicine, Autonomous University of Yucatan, Merida, Mexico. [48]Laboratorio de Patología Veterinaria, Universidad de Caldas, Manizales, Colombia. [49]Centre for Interdisciplinary Research in Animal Health (CIISA), Faculty of Veterinary Medicine, University of Lisbon, Lisboa, Portugal. [50]Four Paws International, Vienna, Austria. [51]Vets Beyond Borders, The Rocks, Australia. [52]Help in Suffering, Jaipur, India. [53]Veterinary Clinic Dr José Rojas, Los Andes, Chile. [54]Department of Biotechnology, Balochistan University of Information Technology, Engineering and Management Sciences, Quetta, Pakistan. [55]Corozal Veterinary Clinic, Corozal Town, Belize. [56]Veterinary Clinic Vetmaster, Ramenskoye, Russia. [57]State Hospital of Veterinary Medicine, Kamianske, Ukraine. [58]Jomo Kenyatta University of Agriculture and Technology, Juja, Kenya. [59]Laboratory of Biomedicine and Regenerative Medicine, Department of Clinical Sciences, Faculty of Animal and Veterinary Sciences, University of Chile, Santiago, Chile. [60]Faculty of Veterinary and Agricultural Sciences, University of Melbourne, Melbourne, Australia. [61]Animal Anti Cruelty League, Port Elizabeth, South Africa. [62]Clinical Sciences Department, Faculty of Veterinary Medicine Bucharest, Bucharest, Romania. [63]Department of Pathology, Faculty of Veterinary Medicine, Ankara University, Ankara, Turkey. [64]Faculty of Veterinary Sciences, National University of Asuncion, San Lorenzo, Paraguay. [65]Lilongwe Society for Protection and Care of Animals (LSPCA), Lilongwe, Malawi. [66]Veterinary clinic Canine Heart, Yelizovo, Russia. [67]MRC-Mitochondrial Biology Unit & Department of Clinical Neurosciences, University of Cambridge, Cambridge, UK. [68]These authors contributed equally: Thomas J. Nicholls, Adrian Baez-Ortega. ✉email: epm27@cam.ac.uk

