## [Peer Review File · Nature Communications]

Reviewers' comments:

Reviewer #1 (Remarks to the Author): Expertise in transmissible cancers

The Strakova et al manuscript provides an in-depth analysis of mitochondrial replacement in the unique transmissible cancer of dogs (CTVT). It was known that mitochondrial replacement has occurred a few times in the evolutionary history of CTVT, but this paper shows that it happened many more times than previously known, and it shows that a single haplotype has been acquired through multiple independent horizontal transfer (HT) events. It is not clear exactly what is driving the selection for the acquisition of this mitochondrial lineage, but it is quite intriguing and may be due to a selfish benefit to the mitogenome itself, rather than providing a selective benefit for the cancer cells which carry it. Overall, the paper has fascinating results which will be of interest to a broad audience in cancer, biology, evolution, and mitochondrial function. The paper's conclusions are sound, and there are only concerns about some issues which could be clarified. The major points which need clarification are the data in Fig 1A/S1/S2 and the data underlying the claim that acquisition of the A1d1a lineage is statistically greater than chance.

Major:

1. Figure 1A presents the main data for the primary conclusion of the paper and it is really too small to interpret. I am not sure of the exact figure size and number limitations, but if it could be made bigger vertically that would be greatly helpful. I understand the full data can be found in the supplementary figure, and this is too large to be a primary figure, but any attempt to make more of the data able to be seen in the main figure would be beneficial. Also, since the primary conclusion of the paper (identification of HT events) is based on these trees, some indication of bootstrap values should be added (probably to the trees in the supplementary data as it would be difficult to add this information into the summary trees in Fig 1A). In particular, there are a pair of HT events (11 and 12, I believe) that are very close together on the tree and also geographically, and the claim that they are separate events is entirely dependent on the confidence in one node. Based on the data provided, there isn't enough data to be sure that these are independent HT events. Also, both HT11 and HT12 are single samples, and they both have two point mutations, but I can't tell from the data provided if those are the same mutations (Perhaps I missed it, but I would like to see some availability of the mitogenome sequences that are the primary data underlying the paper). Also, adding the notation of the number of the HT event to the supplementary tree would greatly aid in understanding.

2. The claim that HT with A1d1a is more common than would be expected by chance is a major finding of the paper and one that underlies the experiments in the second half of the paper. More explanation of the data supporting this claim are needed. On page 26, the authors write that prediction of the expected frequency of HT requires the observed frequency of the haplotypes in global dog populations, but CTVT is not evenly distributed around the world, so they need to control for the prevalence of CTVT in the populations tested (ie. if A1d1a were not very common worldwide, but most cases of CTVT were in populations with A1d1a, then HT would be likely to occur with A1d1a more than would be expected based on the global A1d1a prevalence alone). One of the major factors which may be making this difficult to understand is that it is hard to tell what the control group is. It is listed as a "representative global dog population," (Figure 1B legend) but based on the supplementary tables and later text, it appears that it may be the host dogs which carried the cancer samples used in the study (and in Table S3 legend, it is just referred to as "dog population"). The interpretations of the data are different if it is a control representative dog population or the host dogs themselves. Specifically, if the control group really is the host mitochondria, then it is clear that the dog data are location and population-matched with the cancers analyzed here, but if it is a normal set of non-diseased dogs, then it needs to be made clear how the matching was done to ensure a representative global dog population and what that means. In Table S1, paired cancer and host samples are listed, although it has a few more samples than the 495 listed as controls, and it includes several dogs with no tumour. Are these normal dogs? If so, why are they included? From Table S1 I count 506 host samples and 478 paired samples, so I am not sure of the 495 used for this analysis. It seems that the use of those

cases with both tumour and host data might be the ideal population to use for this analysis. Regardless, further clarification of this analysis is needed.

Minor:

1. The authors show that all the A1d1a HT events are relatively recent, and suggest several hypotheses for this (page 12), but one interesting possibility not mentioned that fits with their overall hypothesis of a selfish element is that CTVT lineages with A1d1a could have occurred in the past, but may have died out over the long term. If the authors think that is reasonable it could be mentioned. Is it known whether this A1d1a haplotype is new within the normal dog population or if it is old?
2. The authors only have three figures in the whole paper and quite a bit of supplementary data. I think some of the data warrant being moved to the main paper. Perhaps Table S2 or Figure S4 could be a part of the main figure. If Figure S4 could include the number of HT events as well as the frequency of A1d1a, then it could aid in visualizing the claim that more HT events than expected were observed.
3. The terminology regarding recombinant genotypes is a bit unclear. In general, it appears the authors are using recombinant to mean a very interesting phenomenon in the two HT3 cases in which the mitogenomes appear to be heteroplasmic with different recombinant genomes present in the same tumour sample. Recombinant mitogenomes can be stable and homoplasmic, so when introducing this it might help to clarify what phenomenon is being described (page 5). Also, when heteroplasmic HT events are described, I am assuming that they are heteroplasmic containing both the ancestral CTVT mtDNA genotype and the new mtDNA genotype, but this is not clear.
4. Page 4. mtDNA capture is defined as mtDNA detectable at ~10% heteroplasmy or more. That is not really the definition of mtDNA capture. It seems that it would be more accurate to say that the authors identified the mtDNA present in the cancers at 10% heteroplasmy or more and then defined mtDNA capture as mtDNA with a phylogenetic relationship that is discordant with the nuclear phylogenetic tree.
5. Page 7, figure S6. Do "dogs with A1d1a" mean normal dogs with A1d1a, diseased dogs with A1d1a in their host tissues, or diseased dogs with A1d1a in their tumours?
6. Page 7. The number of tumour samples with the independent 16660insCC mutations in the HT1 and HT2 backgrounds should be mentioned.
7. Figure 3. Red-green is used to distinguish mitochondrial haplotypes (hard to distinguish for those with colour-blindness, but not an essential change).
8. Page 11. Would it be possible to either experimentally compare the growth rates of CTVT in dogs via experimental infection or retrospectively look at the mitogenotypes of natural infections which have been followed longitudinally in order to test whether A1d1a confers a positive or negative effect on cancer growth? This is clearly not an experiment for this paper, but a possible suggestion for the future which may or may not be feasible.
9. Page 27. The methods for phasing/assembling the mtDNA genomes for HT3 are not explained, and I did not find the sequence data for these inferred phased mitogenomes.
10. Data availability: I did not see any availability of PacBio data or assembled mtDNA genomes.

Reviewer #2 (Remarks to the Author): Expertise in mt HT

This short article describes an intriguing case of repeated horizontal transfer and fixation of a particular mtDNA haplotype (A1d1a) in canine transmissible venereal tumour (CTVT). The authors find a control region polymorphism in A1d1a mtDNA involving a CC dinucleotide insertion. They determine that the effect of this regulatory variant is to reduce transcription of mtDNA-encoded genes, and infer from other data that a decline in mitochondrial transcript abundance is unlikely to be adaptive for CTVT cells. Instead, the authors suggest that A1d1a mtDNA has a replicative advantage that drives the haplotype to fixation (i.e., selfish selection), with the CC-insertion control region favoring replication over transcription.

The overall story here is interesting, provides some novel insights into the properties of the A1d1a

haplotype, and is presented clearly and concisely. I wonder, though, whether there are any experiments that might be done to bolster the inference that replication is favored over transcription in this system? Can replicative intermediates be distinguished and quantified, and compared with those in a CTVT line having a non-A1d1a haplotype? Such data could test the authors' hypothesis which, though certainly reasonable, lacks experimental support at the moment. I realize this may well be asking too much, given that CTVT cell lines are not currently available.

Reviewer #3 (Remarks to the Author): Expertise in mt evolution and transcription

The authors describe the selection for a CC insertion mutation at position 16660 in the control region of mtDNA from a canine transmissible cancer. They demonstrate the recurrence of this mutation in horizontally transferred mtDNA in these tumors, in recombinants with this mutation and the supposedly "wildtype" mtDNA from the tumor, and as spontaneous mutation appearing in a different haplogroup background. The frequency by which this occurs strongly suggests that this mutation is prevalent in these tumors.

The outstanding question remains – how does this mutation lead to a selective advantage for the tumors? Seemingly paradoxically, the authors demonstrate a reduction in the level of mtDNA transcripts that correlates to the insCC. However, it is important to remember that mtDNA transcripts appear to be in a strong excess, as demonstrated in mice with a component of the mt-RNA stability complex knocked out. In these mice, no over phenotype was observed, despite reductions in mt-RNA levels by >50% for most mt-encoded protein genes (PMID: 26247782).

The data presented in this regard look quite convincing, including looking into the preliminaries of mitochondria transcription in the unstudied dog mtDNA. However the main flaw of the manuscript in its current state is that the main question remains unanswered – how does this mutation lead to a strong selection for it in the tumors? The authors speculate that it may be due to the link between LSP transcription and replication, but in the absence of copy number differences, this is mere speculation. The existence of both fixed and heteroplasmic tumors in their samples may actually argue against a strong replicative advantage for the mutation.

Unfortunately, I am not familiar with the biology of this tumor. But is it possible to culture cells from this tumor in the lab? If so, an ethidium bromide / ddC "replication stall" then washout experiment may be able to track the rate of mtDNA recovery in cultured cells bearing and lacking this mutation.

Other than this, my only concern is the idea that is planted in the manuscript of a "selfish" element. The data provided do not in any way suggest that this mutation would be deleterious to the cells, so it is unclear to me why the authors implant the "selfish" idea in the introduction. Either this should be removed or highlighted as an "even selfish elements are known to exist" context. Or a clear statement in the discussion that there is evidence that the insCC mutation is selected for, but whether it is "selfish" remains unclear.

Response to reviewers' comments

Reviewer #1 (Remarks to the Author): Expertise in transmissible cancers

The Strakova et al manuscript provides an in-depth analysis of mitochondrial replacement in the unique transmissible cancer of dogs (CTVT). It was known that mitochondrial replacement has occurred a few times in the evolutionary history of CTVT, but this paper shows that it happened many more times than previously known, and it shows that a single haplotype has been acquired through multiple independent horizontal transfer (HT) events. It is not clear exactly what is driving the selection for the acquisition of this mitochondrial lineage, but it is quite intriguing and may be due to a selfish benefit to the mitogenome itself, rather than providing a selective benefit for the cancer cells which carry it. Overall, the paper has fascinating results which will be of interest to a broad audience in cancer, biology, evolution, and mitochondrial function. The paper's conclusions are sound, and there are only concerns about some issues which could be clarified. The major points which need clarification are the data in Fig 1A/S1/S2 and the data underlying the claim that acquisition of the A1d1a lineage is statistically greater than chance.

Major:

1. Figure 1A presents the main data for the primary conclusion of the paper and it is really too small to interpret. I am not sure of the exact figure size and number limitations, but if it could be made bigger vertically that would be greatly helpful. I understand the full data can be found in the supplementary figure, and this is too large to be a primary figure, but any attempt to make more of the data able to be seen in the main figure would be beneficial.

We have substantially increased the vertical length of Figure 1A and thank the reviewer for this useful suggestion.

Also, since the primary conclusion of the paper (identification of HT events) is based on these trees, some indication of bootstrap values should be added (probably to the trees in the supplementary data as it would be difficult to add this information into the summary trees in Fig 1A).

We added bootstrap values to both the nuclear and mitochondrial phylogenetic trees in the supplementary materials.

In particular, there are a pair of HT events (11 and 12, I believe) that are very close together on the tree and also geographically, and the claim that they are separate events is entirely dependent on the confidence in one node. Based on the data provided, there isn't enough data to be sure that these are independent HT events.

Bootstrap support values show significant confidence in the node separating HT events 11 and 12 in the nuclear tree (bootstrap value = 100), providing strong evidence for these being geographically clustered separate horizontal transfer events. Indeed, there are 97 unique mutations supporting a sister-taxa relationship between samples 1353T (HT11) and 1530T (HT1) with 1539T (HT12) as an outgroup and none supporting a sister-taxa relationship between 1353T (HT11) and 1539T (HT12) with 1530T (HT1) as an outgroup. In addition, there are nine variants that distinguish the A1d1a haplotypes found in these two tumours, most of which (5/9) are germline polymorphisms within the A1d1a haplogroup. This provides additional evidence that the two haplotypes were acquired in separate and independent horizontal transfer events.

Also, both HT11 and HT12 are single samples, and they both have two point mutations, but I can't tell from the data provided if those are the same mutations (Perhaps I missed it, but I would like to see some availability of the mitogenome sequences that are the primary data underlying the paper).

The two potentially somatic variants in 1353T (HT11) and 1539T (HT12) are indeed different (1353T: MT 12567 C>T; MT 14536 A>G and 1539T: MT 6048 C>T; MT 7090 G>A). A summary of mtDNA variants in each sample is now provided in Supplementary Table 2.

Also, adding the notation of the number of the HT event to the supplementary tree would greatly aid in understanding.

We have added the HT event labels to both the nuclear and mtDNA trees and agree that this substantially improves the clarity of these figures.

2. The claim that HT with A1d1a is more common than would be expected by chance is a major finding of the paper and one that underlies the experiments in the second half of the paper. More explanation of the data supporting this claim are needed. On page 26, the authors write that prediction of the expected frequency of HT requires the observed frequency of the haplotypes in global dog populations, but CTVT is not evenly distributed around the world, so they need to control for the prevalence of CTVT in the populations tested (ie. if A1d1a were not very common worldwide, but most cases of CTVT were in populations with A1d1a, then HT would be likely to occur with A1d1a more than would be expected based on the global A1d1a prevalence alone). One of the major factors which may be making this difficult to understand is that it is hard to tell what the control group is. It is listed as a "representative global dog population," (Figure 1B legend) but based on the supplementary tables and later text, it appears that it may be the host dogs which carried the cancer samples used in the study (and in Table S3 legend, it is just referred to as "dog population"). The interpretations of the data are different if it is a control representative dog population or the host dogs themselves. Specifically, if the control group really is the host mitochondria, then it is clear that the dog data are location and population-matched with the cancers analyzed here, but if it is a normal set of non-diseased dogs, then it needs to be made clear how the matching was done to ensure a representative global dog population and what that means. In Table S1, paired cancer and host samples are listed, although it has a few more samples than the 495 listed as controls, and it includes several dogs with no tumour. Are these normal dogs? If so, why are they included? From Table S1 I count 506 host samples and 478 paired samples, so I am not sure of the 495 used for this analysis.

It seems that the use of those cases with both tumour and host data might be the ideal population to use for this analysis. Regardless, further clarification of this analysis is needed.

We thank the reviewer for raising this important point. Of the 'normal' dogs whose sequences are reported in the study, 491/495 were CTVT hosts and 4/495 were dogs unaffected with CTVT. These four unaffected dogs were sequenced unintentionally due to a labelling error, but the sequences generated were nonetheless included as controls.

14/491 CTVT hosts do not have matched tumours included in the study as the tumour sequences generated did not meet quality standards. However, the CTVT host sequences were of good quality and were included in the study as germline controls.

In addition, samples from two unaffected dogs were sequenced multiple times due to a sample labelling error. The duplicated samples were processed as technical controls for our bioinformatics pipelines. These are marked in Supplementary Table 1 and were not double-counted in the sample size tally.

We have clarified these details in Supplementary Table 1 and its legend. In addition, since the large majority of dog germline samples were derived from CTVT hosts, we have now presented this population as 'CTVT host dogs' throughout the text and legends.

Minor:

1. The authors show that all the A1d1a HT events are relatively recent, and suggest several hypotheses for this (page 12), but one interesting possibility not mentioned that fits with their overall hypothesis of a selfish element is that CTVT lineages with A1d1a could have occurred in the past, but may have died out over the long term. If the authors think that is reasonable it could be mentioned. Is it known whether this A1d1a haplotype is new within the normal dog population or if it is old?

This is an interesting possibility, and we have added it as an additional interpretation to the Discussion (page 13, line 392). The time at which A1d1a emerged is not known, but given its worldwide distribution (Supplementary Figure 4), it probably predates the global spread of CTVT.

2. The authors only have three figures in the whole paper and quite a bit of supplementary data. I think some of the data warrant being moved to the main paper. Perhaps Table S2 or Figure S4 could be a part of the main figure. If Figure S4 could include the number of HT events as well as the frequency of A1d1a, then it could aid in visualizing the claim that more HT events than expected were observed.

We agree with the reviewer that moving some of the supplemental data into the main paper may aid clarity. A shortened version of Table S2 has now been added to the main paper. We believe that the higher-than-expected frequency of A1d1a HTs and the apparent geographical clustering of A1d1a HTs is nicely illustrated in Figures 1B and 1C; thus we would argue that moving Supplementary Figure 4 to the main paper may be redundant.

3. The terminology regarding recombinant genotypes is a bit unclear. In general, it appears the authors are using recombinant to mean a very interesting phenomenon in the two HT3 cases in which the mitogenomes appear to be heteroplasmic with different recombinant genomes present in the same tumour sample. Recombinant mitogenomes can be stable and homoplasmic, so when introducing this it might help to clarify what phenomenon is being described (page 5). Also, when heteroplasmic HT events are described, I am assuming that they are heteroplasmic containing both the ancestral CTVT mtDNA genotype and the new mtDNA genotype, but this is not clear.

Yes, the two tumours derived from HT3 each carry heteroplasmic mixtures of recombinant mtDNA products. We have clarified this within the main text (page 5, line 179; page 8, line 260), Figure 2 legend (page 9, line 277), Table 1 legend (page 7, line 229) and Methods (page 27, line 852). We have also clarified that in other cases of heteroplasmy, the heteroplasmy involves the incumbent or parental mtDNA as well as the incoming horizontally transferred haplotype (Figure 1 legend: page 6, line 206; Table 1 legend: page 7, line 226; Methods: page 27, line 852).

4. Page 4. mtDNA capture is defined as mtDNA detectable at ~10% heteroplasmy or more. That is not really the definition of mtDNA capture. It seems that it would be more accurate to say that the authors identified the mtDNA present in the cancers at 10% heteroplasmy or more and then defined mtDNA capture as mtDNA with a phylogenetic relationship that is discordant with the nuclear phylogenetic tree.

We thank the reviewer for this astute comment. We have clarified the definition of horizontal transfer inference (page 27, line 848).

5. Page 7, figure S6. Do “dogs with A1d1a” mean normal dogs with A1d1a, diseased dogs with A1d1a in their host tissues, or diseased dogs with A1d1a in their tumours?

We have clarified this point in the main text (page 8, line 238). We have also clarified the nature of the ‘normal dog’ population throughout the paper (see response to major comment point 2 above).

6. Page 7. The number of tumour samples with the independent 16660insCC mutations in the HT1 and HT2 backgrounds should be mentioned.

This has now been clarified (page 8, line 263).

7. Figure 3. Red-green is used to distinguish mitochondrial haplotypes (hard to distinguish for those with colour-blindness, but not an essential change).

There is a limit of colours, and red and green provide contrast and are commonly used primary colours. However, we appreciate that this might cause difficulty for colour-blind readers, and we would be happy to alter this if required by the editor.

8. Page 11. Would it be possible to either experimentally compare the growth rates of CTVT in dogs via experimental infection or retrospectively look at the mitogenotypes of natural infections which have been followed longitudinally in order to test whether A1d1a confers a positive or negative effect on cancer growth? This is clearly not an experiment for this paper, but a possible suggestion for the future which may or may not be feasible.

We have in fact partially addressed the question as to whether A1d1a may have an adaptive selective advantage by assessing the frequencies of mitotic figures in CTVTs with different mtDNA haplotypes (Supplementary Figure 8A) and by measuring time taken to respond to chemotherapy (Supplementary Figure 8B). We did not detect differences in these traits between A1d1a CTVTs and those with other haplotypes, which is part of our argument that the A1d1a selective advantage may not be adaptive, but rather ‘selfish’.

9. Page 27. The methods for phasing/assembling the mtDNA genomes for HT3 are not explained, and I did not find the sequence data for these inferred phased mitogenomes.

Phasing of recombinant mtDNA haplotypes was performed manually by visualising long reads using a genome browser. This has been clarified in the methods (page 29, line 921). PacBio data have been

uploaded to ENA, with accession information added to the Data Availability section (page 32, line 1039).

10. Data availability: I did not see any availability of PacBio data or assembled mtDNA genomes.

PacBio data have been uploaded to ENA, with accession information added to the Data Availability section (page 32, line 1039), and will be publicly released upon publication. We have added a supplementary file listing genotypes analysed in this paper (Supplementary Table 2).

Reviewer #2 (Remarks to the Author): Expertise in mt HT

This short article describes an intriguing case of repeated horizontal transfer and fixation of a particular mtDNA haplotype (A1d1a) in canine transmissible venereal tumour (CTVT). The authors find a control region polymorphism in A1d1a mtDNA involving a CC dinucleotide insertion. They determine that the effect of this regulatory variant is to reduce transcription of mtDNA-encoded genes, and infer from other data that a decline in mitochondrial transcript abundance is unlikely to be adaptive for CTVT cells. Instead, the authors suggest that A1d1a mtDNA has a replicative advantage that drives the haplotype to fixation (i.e., selfish selection), with the CC-insertion control region favoring replication over transcription.

The overall story here is interesting, provides some novel insights into the properties of the A1d1a haplotype, and is presented clearly and concisely. I wonder, though, whether there are any experiments that might be done to bolster the inference that replication is favored over transcription in this system? Can replicative intermediates be distinguished and quantified, and compared with those in a CTVT line having a non-A1d1a haplotype? Such data could test the authors' hypothesis which, though certainly reasonable, lacks experimental support at the moment. I realize this may well be asking too much, given that CTVT cell lines are not currently available.

We thank the reviewer for their comments and are pleased that they found the paper interesting and intriguing. We agree that additional data supporting our hypothesis that A1d1a favours replication over transcription and is thus favoured via 'selfish' selection could strengthen our conclusions. In response to the reviewer's suggestion, we attempted to identify and quantify the oriH primer, a putative replication intermediate originating at the mtDNA light strand promoter (LSP). This primer has not been annotated in the canine genome and an attempt to characterise its boundaries using RNAseq was not successful; nevertheless we proceeded with the analysis by defining a region where the primer may lie based on analogy to human mtDNA. We mapped RNAseq reads from various CTVT haplotypes to the defined region and saw no clear difference in putative primer transcript abundance between groups. However, due to the lack of positive controls and the fact that such a method has not previously been described for this purpose (even in humans and model systems), we were not confident that the assay was capable of identifying differences in replication activity between haplotypes. This analysis is also confounded by the fact that, at least in the human mitochondrial transcription system, transcription initiation from LSP generates both the replication primer and a short polyadenylated RNA species known as 7S RNA which is of very similar length. The relationship between the rate of 7S RNA synthesis and the rate of replication primer synthesis remains unclear, and so is problematic as a readout for the rate of replication primer synthesis.

We also attempted to establish an *in vitro* transcription assay in order to further explore the effect of 16660insCC on transcription initiation and replication primer production. However, after some pilot experiments, it became apparent that this would require the expression and purification of a complete set of recombinant canine mitochondrial transcription factors, a task that was not feasible within the time-frame of this study. Additionally, as noted by the reviewer, the lack of CTVT cell lines is a major obstacle in the experimental investigation of this cancer's mitochondrial biology. Establishment of CTVT cell lines is an ongoing priority within our laboratory, but one that is unlikely to yield results within a reasonable time period. Were cell lines to become available we would have the ability to study the pattern of mtDNA replication intermediates using 2D agarose gel electrophoresis, although it is also unclear whether the resolution of this method would be sufficient to detect differences between the different haplotypes.

Overall, we accept that one of the limitations of our manuscript is the lack of direct experimental measurement of A1d1a replication activity. However, without substantial investment in reagent development (which would take years to complete), we do not see how this will be possible within the context of the current study.

Reviewer #3 (Remarks to the Author): Expertise in mt evolution and transcription

The authors describe the selection for a CC insertion mutation at position 16660 in the control region of mtDNA from a canine transmissible cancer. They demonstrate the recurrence of this mutation in horizontally transferred mtDNA in these tumors, in recombinants with this mutation and the supposedly "wildtype" mtDNA from the tumor, and as spontaneous mutation appearing in a different haplogroup background. The frequency by which this occurs strongly suggests that this mutation is prevalent in these tumors.

The outstanding question remains – how does this mutation lead to a selective advantage for the tumors? Seemingly paradoxically, the authors demonstrate a reduction in the level of mtDNA transcripts that correlates to the insCC. However, it is important to remember that mtDNA transcripts appear to be in a strong excess, as demonstrated in mice with a component of the mt-RNA stability complex knocked out. In these mice, no over phenotype was observed, despite reductions in mt-RNA levels by >50% for most mt-encoded protein genes (PMID: 26247782).

We thank the reviewer for alerting us to this interesting study. It is certainly possible that the relatively modest reduction in mtDNA protein-coding transcript abundance that we observed in A1d1a CTVTs has no effect on the phenotype of the cell. Indeed, dogs with germline A1d1a have no overt phenotype. We have added a comment to the text noting that the decreased transcript abundance observed in A1d1a CTVT may have no phenotype, citing the relevant study (page 11, line 327).

The data presented in this regard look quite convincing, including looking into the preliminaries of mitochondria transcription in the unstudied dog mtDNA. However the main flaw of the manuscript in its current state is that the main question remains unanswered – how does this mutation lead to a strong selection for it in the tumors? The authors speculate that it may be due to the link between LSP transcription and replication, but in the absence of copy number differences, this is mere speculation. The existence of both fixed and heteroplasmic tumors in their samples may actually argue against a strong replicative advantage for the mutation.

We are glad that the reviewer finds the data convincing, and agree that the lack of direct experimental evidence linking 16660insCC with mtDNA replication is a limitation of the study (see comments to Reviewer 2).

The reviewer raises a valid point about heteroplasmy. To clarify, although we did identify heteroplasmy following mtDNA horizontal transfer involving some haplotypes, all but one A1d1a horizontal transfers were homoplasmic. The only A1d1a exception was HT3, which carried a heteroplasmic mixture of HT1/A1d1a haplotypes fixed for 16660insCC. Thus, in every case of A1d1a horizontal transfer, 16660insCC was fixed. The occurrence of heteroplasmy, and the haplotypes with which it is associated, are now highlighted in Table 1.

Unfortunately, I am not familiar with the biology of this tumor. But is it possible to culture cells from this tumor in the lab? If so, an ethidium bromide / ddC “replication stall” then washout experiment may be able to track the rate of mtDNA recovery in cultured cells bearing and lacking this mutation.

A mitochondrial depletion and recovery experiment would be an excellent method to investigate the replication potential of A1d1a relative to other haplotypes. Unfortunately, however, the unavailability of CTVT cell lines, or of isogenic dog cell lines carrying different mtDNA haplotypes, precludes this. The development of these reagents is outside of the scope of this study (see comment to Reviewer 2, above).

Other than this, my only concern is the idea that is planted in the manuscript of a “selfish” element. The data provided do not in any way suggest that this mutation would be deleterious to the cells, so it is unclear to me why the authors implant the “selfish” idea in the introduction. Either this should be removed or highlighted as an “even selfish elements are known to exist” context. Or a clear statement in the discussion that there is evidence that the insCC mutation is selected for, but whether it is “selfish” remains unclear.

‘Selfish’ selection is, according to our definition, selection that drives change in the frequency of a haplotype due to altered replication or segregation potential, regardless of the effect on the host cell. This definition is stated in the introduction (page 4, line 138) and we believe that it is widely accepted by the community (see for example Klucnika and Ma, *Open Biology* 2019).

REVIEWERS' COMMENTS:

Reviewer #1 (Remarks to the Author):

Overall the revised version of the Strakova et al manuscript answers nearly all reviewer concerns. This is an exciting study and will be of broad interest. The finding of repeated mitochondrial transfer and selective transfer of a single haplotype are clear, and the data suggesting that a single CC insertion is responsible are strong, and the implications are fascinating. The suggestion that it may be due to selfish mechanisms is interesting and agrees with the data. The authors do not overreach in this claim, and more conclusive research that will test this hypothesis in the future are well beyond the scope of this manuscript. I have one final concern, which I believe should be a relatively simple change in the data analysis of one part of one figure, and one very minor comment, which is likely just a typo. If the correction has no change in the statistical significance of the claim (which seems likely), then I have no further concerns for this paper and look forward to seeing it published.

Based on the explanation given in the rebuttal and in the supplementary data, it seems to me that Figure 1B and the statistical analysis it described should only include the 477 host dogs with CTVT matched samples if the intent is to compare the frequency of haplotypes between a representative sample of dogs and with the number of CTVT HT events. It appears the CTVT samples which represent HT5 and HT13 do not have matching host tissue, so they could be excluded. In this way the populations of the control group would match the population out of which all the HT events were selected. Alternatively, mtDNA from normal animals from the same population could be used to be a match for samples with no matching host sample, but that would entail further sequencing-restricting this specific analysis to only matched host/CTVT samples would be the easiest solution to this issue. Including other animals from other populations with no CTVT or including CTVT with no matched host makes for a non-representative control group and calls into question the comparison, even if those data are of high quality. In the legend it can be stated that this is the subset of cases for which both host and CTVT mtDNA sequence are available. Using all 539 CTVT samples that passed quality filters is appropriate in Figure 1 part A, C, and D. I agree that this control group should be referred to as CTVT host dogs, but only if the non-affected dogs are removed. If the authors do not want to change this figure panel and wish to display all the available samples here, then the analysis with only samples for which matched genotypes are available (477 host dogs and 17 HT events by my count) should at least be done and the statistical significance should be reported in the text. The data are available in the supplementary tables and an additional supplementary figure would not be necessary. This will most likely not change the conclusion that the frequency of A1d1a in host dogs is lower than that in CTVT HT events, but it should be done in order to be a valid comparison, and it does not involve acquisition of any additional sequence data.

Page 10, line 316. Text says three tumours, but Supplementary Table 5A says two. Whichever is wrong should be corrected. I believe two was listed earlier in the text as well (Line 267), so I suspect two is correct.

Reviewer #3 (Remarks to the Author):

My only suggestion - the use of cells as an ex vivo assay to test the molecular details of the mutation - appears to be technically unfeasible due to the inaccessibility of the cells. As such, I see no further reason to delay publication.

I hope that this system is further developed for future research, pending the approval by the other reviewers.

Responses to Reviewers' Comments for manuscript NCOMMS-19-36441, Strakova et al.

REVIEWERS' COMMENTS:

Reviewer #1 (Remarks to the Author):

Overall the revised version of the Strakova et al manuscript answers nearly all reviewer concerns. This is an exciting study and will be of broad interest. The finding of repeated mitochondrial transfer and selective transfer of a single haplotype are clear, and the data suggesting that a single CC insertion is responsible are strong, and the implications are fascinating. The suggestion that it may be due to selfish mechanisms is interesting and agrees with the data. The authors do not overreach in this claim, and more conclusive research that will test this hypothesis in the future are well beyond the scope of this manuscript. I have one final concern, which I believe should be a relatively simple change in the data analysis of one part of one figure, and one very minor comment, which is likely just a typo. If the correction has no change in the statistical significance of the claim (which seems likely), then I have no further concerns for this paper and look forward to seeing it published.

Based on the explanation given in the rebuttal and in the supplementary data, it seems to me that Figure 1B and the statistical analysis it described should only include the 477 host dogs with CTVT matched samples if the intent is to compare the frequency of haplotypes between a representative sample of dogs and with the number of CTVT HT events. It appears the CTVT samples which represent HT5 and HT13 do not have matching host tissue, so they could be excluded. In this way the populations of the control group would match the population out of which all the HT events were selected. Alternatively, mtDNA from normal animals from the same population could be used to be a match for samples with no matching host sample, but that would entail further sequencing--restricting this specific analysis to only matched host/CTVT samples would be the easiest solution to this issue. Including other animals from other populations with no CTVT or including CTVT with no matched host makes for a non-representative control group and calls into question the comparison, even if those data are of high quality. In the legend it can be stated that this is the subset of cases for which both host and CTVT mtDNA sequence are available. Using all 539 CTVT samples that passed quality filters is appropriate in Figure 1 part A, C, and D. I agree that this control group should be referred to as CTVT host dogs, but only if the non-affected dogs are removed. If the authors do not want to change this figure panel and wish to display all the available samples here, then the analysis with only samples for which matched genotypes are available (477 host dogs and 17 HT events by my count) should at least be done and the statistical significance should be reported in the text. The data are available in the supplementary tables and an additional supplementary figure would not be necessary. This will most likely not change the conclusion that the frequency of A1d1a in host dogs is lower than that in CTVT HT events, but it should be done in order to be a valid comparison, and it does not involve acquisition of any additional sequence data.

We thank the reviewer for their comment regarding the representativeness of the tumour and normal host dog populations in Figure 1B. We performed the analysis that the reviewer suggested, restricting the analysis to include only matched tumour-host pairs (477 host dogs and 17 horizontal transfer events). Restricting the analysis to only matched tumour-host pairs had no effect on the level of statistical significance ($p < 0.001$). We have added a comment to the text to clarify this.

Page 10, line 316. Text says three tumours, but Supplementary Table 5A says two. Whichever is wrong should be corrected. I believe two was listed earlier in the text as well (Line 267), so I suspect two is correct.

We have rephrased the manuscript text to clarify the number of tumours with 16660insCC (two) and 16660insC (one). This information is also detailed in Supplementary Table 2C.

Reviewer #3 (Remarks to the Author):

My only suggestion - the use of cells as an ex vivo assay to test the molecular details of the mutation - appears to be technically unfeasible due to the inaccessibility of the cells. As such, I see no further reason to delay publication.

I hope that this system is further developed for future research, pending the approval by the other reviewers.